

# The regulatory genome constrains protein sequence evolution: implications for the search for disease-associated genes

Patrick Evans[1], Nancy J. Cox[1] and Eric R. Gamazon[1,2,3,4]

[1] Division of Genetic Medicine, Vanderbilt University Medical Center, Nashville, TN, United States of America
[2] Clare Hall, University of Cambridge, Cambridge, United Kingdom
[3] MRC Epidemiology Unit, University of Cambridge, Cambridge, United Kingdom
[4] Data Science Institute, Vanderbilt University, Nashville, TN, United States of America

Corresponding authors
Patrick Evans,
patrick.evans@vanderbilt.edu,
eric.gamazon@vanderbilt.edu
Eric R. Gamazon,
eric.gamazon@vanderbilt.edu

## ABSTRACT

The development of explanatory models of protein sequence evolution has broad implications for our understanding of cellular biology, population history, and disease etiology. Here we analyze the GTEx transcriptome resource to quantify the effect of the transcriptome on protein sequence evolution in a multi-tissue framework. We find substantial variation among the central nervous system tissues in the effect of expression variance on evolutionary rate, with highly variable genes in the cortex showing significantly greater purifying selection than highly variable genes in subcortical regions (Mann–Whitney U $p = 1.4 \times 10^{-4}$). The remaining tissues cluster in observed expression correlation with evolutionary rate, enabling evolutionary analysis of genes in diverse physiological systems, including digestive, reproductive, and immune systems. Importantly, the tissue in which a gene attains its maximum expression variance significantly varies ($p = 5.55 \times 10^{-284}$) with evolutionary rate, suggesting a tissue-anchored model of protein sequence evolution. Using a large-scale reference resource, we show that the tissue-anchored model provides a transcriptome-based approach to predicting the primary affected tissue of developmental disorders. Using gradient boosted regression trees to model evolutionary rate under a range of model parameters, selected features explain up to 62% of the variation in evolutionary rate and provide additional support for the tissue model. Finally, we investigate several methodological implications, including the importance of evolutionary-rate-aware gene expression imputation models using genetic data for improved search for disease-associated genes in transcriptome-wide association studies. Collectively, this study presents a comprehensive transcriptome-based analysis of a range of factors that may constrain molecular evolution and proposes a novel framework for the study of gene function and disease mechanism.

# INTRODUCTION

Protein sequence evolution is a central concern for the fields of molecular biology and comparative genomics. Indeed, characterizing the determinants of the rate of protein

evolution may help to clarify a wide range of biological processes and phenomena, including the molecular basis of adaptation, the evolution of speciation, and the genetic etiology of disease. Although numerous studies have proposed possible determinants (*Drummond et al., 2005*; *Drummond & Wilke, 2008*; *Yang, Zhuang & Zhang, 2010*; *Yang et al., 2012*), the underlying mechanisms and potential interactions remain unclear.

Protein evolution reflects both the rate at which new nucleotide mutations arise and the rate of repair. A second engine of protein evolution, fixation of new mutations depends upon the fitness effect of the relevant mutations and the balance between selection and genetic drift, which is in part mediated by effective population size. We can quantify selection pressures acting on protein-coding regions using the widely used dN/dS ratio. The ratio compares the rate of substitutions ($K_a$ or dN) at nonsynonymous sites, which are presumed to undergo selection, to the rate of substitutions ($K_s$ or dS) at synonymous sites, which are presumed neutral so that dS may serve as a proxy for mutation rate (*Kimura, 1990*) (although selection may operate on silent sites to favor certain codons (*Akashi, 2001*; *Plotkin & Kudla, 2011*; *Quax et al., 2015*). The ratio provides evidence for selective constraint if dN/dS is significantly less than 1, for neutrality if dN/dS equals 1, or for positive selection if dN/dS is significantly greater than 1. Requiring dN/dS to be significantly greater than one is generally conservative, as positive evolution generally acts on one region or domain of a protein while the rest of the protein evolves under purifying selection; hence dN/dS may potentially overlook positions under positive selection. Here we use the ratio dN/dS as a measure of protein "evolutionary rate," as previously described (*Zhang & Yang, 2015*). The rate of protein evolution may vary greatly within and between species (*Duret & Mouchiroud, 2000*; *Rocha & Danchin, 2004*; *Drummond et al., 2005*; *Larracuente et al., 2008*), and elucidating the causes of this variation is an important question in molecular evolution.

Genetic mutations typically alter phenotype either by altering proteins or by affecting their regulation. Indeed, differential regulation of gene expression is thought to underlie the remarkable divergence in traits (e.g., behavioral) between humans and chimpanzees especially given the high degree of similarity between orthologous protein sequences (*King & Wilson, 1975*). Much of the early work was performed in single cell organisms, such as yeast, raising the question of the generalizability of the results to multicellular eukaryotes (*Pál, Papp & Hurst, 2001*; *Akashi, 2003*; *Drummond, Raval & Wilke, 2006*). Different tissues and cell types express different genes and the regulation of gene expression may vary substantially across tissues and cell types, strongly influencing organism-level traits and modulating the evolution of phenotypic novelty. Furthermore, the expression profile of some genes may exhibit substantial developmental-stage specificity.

Molecular evolution in coding sequences has been investigated in many taxa, including humans, and reported to be determined by several (potentially mutually correlated) factors, including expression level, expression breadth, recombination rate, robustness to mistranslation, and connectivity in a biophysical network (*Duret & Mouchiroud, 2000*; *Pál, Papp & Hurst, 2001*; *Fraser et al., 2002*; *Jordan et al., 2002*; *Wall et al., 2005*; *Drummond et al., 2005*; *Larracuente et al., 2008*; *Drummond & Wilke, 2008*; *Park & Choi, 2010*; *Hudson & Conant, 2011*; *Shen et al., 2011*; *Kryuchkova-Mostacci & Robinson-Rechavi, 2015*). Despite

the complexity and spatiotemporality of gene expression in a multicellular organism (with its long lifespan and developmental processes), the expression level of a gene has been shown to be a primary predictor of the rate of protein sequence evolution with a variety of possible underlying mechanisms proposed (*Akashi, 2003*; *Rocha & Danchin, 2004*; *Lemos, Meiklejohn & Hartl, 2004*; *Drummond et al., 2005*). Expression breadth has also been shown to affect protein evolutionary rate, though expression level and breadth have a strong correlation (*Duret & Mouchiroud, 2000*; *Zhang & Li, 2004*; *Liao, Scott & Zhang, 2006*; *Park et al., 2012*). However, studies of the relationship between gene expression and protein evolution have been limited to divergence data for several hundred to thousand genes, and to expression data measured in a small number of tissues (*Duret & Mouchiroud, 2000*; *Wagner, 2005*) using microarrays.

There has been an explosion of large-scale genomic and other types of omics data in a variety of tissue and cellular contexts (*Pickrell et al., 2010*; *GTEx Consortium, 2013*; *Lappalainen et al., 2013*), motivating our attempt at an integrated view of the evolutionary signature of known genes. Here we report a comprehensive transcriptome-based analysis of the factors that may constrain the rate of protein evolution and examine their relevance to sets of genes that define a spectrum of clinical and disease effects, from one end, the essential genes, to the other end, the loss-of-function (LOF) tolerant genes (*Ashburner et al., 2000*; *Blair et al., 2013*). Notably, we examine several methodological implications on genomic approaches to mapping disease-relevant genes and on the study of gene function.

## MATERIALS & METHODS

### Data

We utilized evolutionary conservation scores from BioMart (*Smedley et al., 2015*) for human-chimp and human-mouse comparisons. In our analyses, we collected synonymous (dS) and nonsynonymous (dN) substitution rates for 23,816 genes from these comparisons. Proteins vary by two to three orders of magnitude in their rate of evolution (dN/dS).

We utilized GTEx v6p release expression data for 53 tissues involving 8,555 RNA-Seq samples. The mean RPKM and median RPKM were calculated for each gene in each tissue. The variance and ratio of variance to mean were also calculated for each gene in each tissue to estimate inter-individual transcriptional variability (see sample size for each tissue in Table S1). For each gene, we identified the tissue in which the gene shows maximum expression variance. A list of the 53 tissues with information on sample size is in Table S1. In addition to the multi-region sampling of the brain, "frontal cortex" and "cerebellar hemisphere" (obtained after receipt by the brain bank) were sampled by the GTEx Consortium in duplicate ("cortex" and "cerebellum," respectively, as the first replicate obtained at initial tissue collection). Availability of lymphoblastoid cell lines (LCLs, derived from blood) and cultured primary fibroblasts (from skin) provided an opportunity to compare the expression variance and network constraints (see below) in the cell types with those in the tissues of origin.

For all correlation analyses presented, we used log2-transformed RPKM values. We calculated the Spearman's $\rho$ and the corresponding $p$-value with evolutionary rate for all

predictors using the R statistical software. To fit a nonlinear model (e.g., of the effect of expression on evolutionary rate), we assumed a generalized additive model using cubic regression spline as "smooth function."

To test for the robustness of our results to technical and experimental confounding, we calculated the Probabilistic Estimation of Expression Residuals (PEER) (*Stegle et al., 2012*) factors (which are based on Factor Analysis) and used the residual to test the correlation of expression level and expression variability with evolutionary rate. We compared results across all tissues (Spearman correlation) between pre-PEER and post-PEER analysis.

To quantify tissue expression breadth, we applied the tau ($\tau$) statistic to the 53 GTEx tissues, with the exception that the frontal cortex was the only brain region used so that the highly correlated tissue samples from the brain would not bias the estimate (*Yanai et al., 2005*; *Kryuchkova-Mostacci & Robinson-Rechavi, 2016*).

$$\tau = \sum_{i=1}^{n} \frac{1 - \widehat{x_i}}{n - 1}$$

where

$$\widehat{x_i} = x_i / \max_{1 \leq j \leq n} \{x_j\}$$

and $x_i$ provides the expression value for the gene in tissue $i$, and $n$ is the number of tissues. We note that $\tau$ yields a score between zero and one, with zero indicating the same expression across tissues and a value of one indicating strong tissue-specific expression. Genes with similarly high levels or similarly low levels across tissues would have high "expression breadth," which here thus refers to the extent of similarity across tissues. In the actual data, $\tau$ attains a minimum of 0.204 and median of 0.82 for protein-coding genes, underscoring the high tissue specificity of a large set of protein-coding genes. Thus, $\tau$ also reflects inter-tissue variability (to be distinguished from inter-individual variability in each tissue) in the expression of a gene.

We utilized protein network data in *Homo sapiens* obtained from STRING v10 (*Szklarczyk et al., 2015*) to investigate the correlation of the number of interactions with evolutionary rate and expression.

We examined several subsets of genes that define a spectrum of phenotypic effects. We obtained and curated a compendium of Mendelian disease genes from the Online Mendelian Inheritance in Man (OMIM). Essential genes were obtained from a study (*Georgi, Voight & Bućan, 2013*) that identified human orthologs of mouse genes with known lethal phenotype from the Mouse Genome Database (*Blake et al., 2011*). LOF-tolerant genes, which can be inactivated without obvious clinical effect, were collected from a comprehensive survey of LOF variants in protein-coding genes (*MacArthur et al., 2012*). We also investigated immune response genes and olfactory genes, as annotated by the Gene Ontology Consortium (*Gene Ontology Consortium, 2015*).

## Tissue-anchored model of evolutionary rate

We identified the tissue (MaxTissue) in which a gene attains its maximum inter-individual expression variance (MaxVariance)—and, thus, perhaps the tissue in which the gene

exhibits its full range of functional activity—and tested the MaxTissue's association with evolutionary rate. We should emphasize that the MaxTissue is not necessarily the only tissue in which a gene functions, and indeed a gene may have multiple functions in different tissues. A rejection of the null hypothesis would indicate support for the hypothesis (i.e., the "tissue-anchored model") that when genes are classified into their MaxTissues, the variance $\sigma^2_{across}$ in evolutionary rate across tissues is greater than the variance $\sigma^2_{within}$ of the tissues.

The MaxTissue's association with evolutionary rate was tested using the non-parametric Kruskal–Wallis test. To determine whether rejection of the null hypothesis is driven by a single outlier tissue or a class of tissues, we calculated the median evolutionary rate and a metric, MaxTissue-$\delta$, for each MaxTissue, defined as

$$3.14(v_{75} - v_{25})/\sqrt{n}$$

where $v_{75}$ and $v_{25}$ are the 75th and 25th percentile for evolutionary rate respectively and $n$ is the number of genes. We also performed empirical analysis by permutation-based tissue assignment ($N = 1000$), preserving the gene count in the observed MaxTissue configuration, for the entire protein-coding set with available human-mouse data on rates of nonsynonymous and synonymous substitutions. For each permutation of the entire set (RandomTissue), the $H$-statistic from the Kruskal–Wallis test was calculated. The empirical $p$-value was the proportion of the total number of permutations in which the RandomTissue $H$-statistic matched or exceeded the observed value from the MaxTissue assignment.

We can define a novel evolutionary signature, the "evolutionary rate of a tissue," as the median evolutionary rate of the genes for which the tissue is a MaxTissue. MaxTissue-$\delta$ is thus a kind of "confidence interval" for this evolutionary signature. Under the tissue-anchored model, there is a statistically significant difference between tissues for this evolutionary signature. (We note that this signature is to be distinguished from the median evolutionary rate of all expressed genes in a tissue).

The tissue-anchored model starts from the observation that the correlation between within-tissue features (e.g., expression level) and evolutionary rate significantly varies by tissue, raising the centrality of a cross-tissue analysis. The model would suggest that variation between developmental programs in which transcription and other within-tissue features exert their effect may be contributing to variation in evolutionary rate. In each MaxTissue, we ranked the genes according to expression variance from high to low. We then identified the significant Gene Ontology biological processes (Benjamini–Hochberg FDR $< 0.05$) for the top genes ($N = 100$) in a MaxTissue. The presence of significant, non-overlapping biological processes between MaxTissues with significant difference in median evolutionary rate (Kruskal–Wallis test) would suggest the importance of developmental processes for constraining evolutionary rate.

## Tissue-anchored model: constraint of developmental programs on evolutionary rate

To further investigate the physiological and developmental mechanisms underlying the tissue-anchored model of evolutionary rate, we sought to determine to what extent

the MaxTissue for a gene could predict the primary affected tissue of developmental disorders associated with the gene. We utilized a unique UK-wide collaborative resource (generated by over 180 clinicians across 24 regional genetics services), the Deciphering Developmental Disorders (DDD) study (*Martin et al., 2018*) which has conducted genome-wide genotyping and whole-exome sequencing of children with developmental disorders and their parents. This resource provides a list of genes associated with developmental disorders as well as affected organs. For each gene, we considered the significance of the overlap of MaxTissue with the primary affected organ (using a chi-square test of the null hypothesis of the independence of the MaxTissue and affected tissue).

## Testing for independent effects given noisy omics data and estimation of spurious correlation

Partial rank correlation analysis of omics data between two variables, $D$ (a determinant) and $K$ (evolutionary rate), while controlling for a third variable $X$ may generate spurious results (*Liu, 1988*; *Drummond, Raval & Wilke, 2006*). Consider the case in which a noisy version $X'$ is distributed with mean equal to $X$ and variance equal to $\sigma_{X'}^2$:

$$X' \sim \left(X, \sigma_{X'}^2\right).$$

We assume that $D$ and $K$ have the following distributions: $D \sim \left(X, \sigma_D^2\right)$ and $K \sim \left(X, \sigma_K^2\right)$. Then the partial rank correlation $r_{DK|X'}$ between $D$ and $K$ given $X'$ simplifies to the following expression:

$$r_{DK|X'} = \frac{\sigma_{X'}^2}{f(\sigma_D^2, \sigma_K^2, \sigma_{X'}^2)}.$$

Here the denominator is some function of all the variances. Importantly, the numerator is nonzero in the presence of noise in $X$ even under the null hypothesis (i.e., $r_{DK|X} = 0$), leading to a spurious correlation. To determine whether the determinant $D$ contributes to $K$ after controlling for $X$ (given that conventional partial rank correlation analysis generates spurious correlations when applied to noisy biological data), we performed permutation analysis, in which $D$ and $X'$ were shuffled together (thus preserving their correlation) $n$ times (here $n = 1,000$) and the partial rank correlation between $D$ and $K$ was assessed within each such permutation null set. This generates an empirical $p$-value for the significance of the observed nonzero correlation coefficient (as the proportion of the permutation null sets that match or exceed the observed correlation coefficient) as well as quantifies the magnitude of the spurious correlation. The permutation null distribution $M_0$ for $r_{DK|X'}$ can be used to estimate an adjusted partial rank correlation coefficient:

$$\widehat{\vartheta_{DK|X}} = \sqrt{\widehat{r_{DK|X'}^2} - (\widehat{E(M_0)})^2}$$

where $\widehat{r_{DK|X'}}$ is the observed partial rank correlation coefficient and $E(.)$ is the expectation operator. We call this approach Empirical Partial Rank Correlation Analysis (EPRCA), which facilitates a test for independent effects on evolutionary rate. For example, since expression level and tissue breadth were found to be correlated, we tested their independent

effect on evolutionary rate using this approach. The approach, in addition, provides an estimate of the extent of the spurious correlation (from the mean and standard deviation of the permutation null distribution).

## Correlation between expression and evolutionary rate given co-expression

The correlation in gene expression (for genes that belong to the same co-expression network) may bias our estimate of the correlation between gene expression and evolutionary rate. We therefore fit a model using Generalized Least Squares (GLS) to account for the non-independence of genes in a co-expression network (shown here, for convenience, for the nonsynonymous substitution rate, but our approach extends, without loss of generality, to dN/dS):

$$G = \pi \, dN + \varepsilon$$

$$var(\varepsilon) = \sigma^2 \Delta$$

where $dN$ is the vector of nonsynonymous substitution rates for a set of genes, $G$ is a vector of gene expression, $\pi$ is the effect size, $\Delta$ is the (known) gene expression covariance matrix, and $\sigma^2$ is the unknown (absolute) scale. The GLS effect size estimate solves the following minimization problem:

$$\hat{\pi} = \underset{\pi}{\mathrm{argmin}}(G - \pi \, dN)^T \Delta^{-1}(G - \pi \, dN)$$

which implies:

$$\hat{\pi} = (dN^T \Delta^{-1} dN)^{-1} dN^T \Delta^{-1} G$$

$$var(\hat{\pi}) = (dN^T \Delta^{-1} dN)^{-1} \sigma^2.$$

We can view the gene expression traits as mapping to a phylogenetic tree such that the covariance matrix $\Delta$ captures the covariance between each pair of tips in the tree. The GLS model implements regression that accounts for the phylogeny.

## Null phylogenies and null networks

To assess the significance of the correlation between amino acid substitution rate and expression while adjusting for branch assignment (defined in Supplementary Information), we shuffled the evolutionary rate estimates for genes while preserving gene branch assignment within the vertebrate phylogeny. Furthermore, we implemented a node degree-preserving permutation method to generate null gene networks, which were used to control for potential bias that may affect the correlation between evolutionary rate and expression. For branch assignment or node degree, we calculated the empirical $p$-value as the proportion of phylogeny- or degree- preserving permutations (out of 10,000), respectively, for which the permutation correlation test statistic matched or exceeded the observed statistic in the actual data.
### Confounding due to tissue diversity sampling

Despite the comprehensiveness of the GTEx resource, the collection of tissues examined here is still only a partial subset of all human tissues and each tissue is composed of many cell types. We therefore evaluated the robustness to tissue sampling of the correlation between $\tau$ and evolutionary rate by calculating the correlation on each of 100 random subsets of 10 and 30 tissues (selected from the 44 tissues).

### Cis heritability of gene expression

We estimated the proportion of gene expression variance captured by local genetic variation and quantified its contribution to evolutionary rate. We considered the following linear mixed model:

$$Y = W + T + \varepsilon$$

$$var(Y) = A\sigma^2 + B\sigma_T^2 + I\sigma_\varepsilon^2.$$

where $Y$ is the residual gene expression phenotype $n$-dimensional vector after adjusting for hidden factors (with $n$ equal to the number of samples in the reference transcriptome dataset), $W$ is the polygenic *cis* contribution to gene expression for SNPs within 1 Mb of the gene, $A$ is the genetic relatedness matrix estimated from the local polymorphism data (*Yang et al., 2010*), $T$ is the polygenic *trans* contribution from the remaining common variants (MAF > 0.05) in the genome, and $B$ is the genetic relatedness matrix estimated from these trans-variants on the other chromosomes. The variance of the polygenic *cis* burden $W$ is $A\sigma^2$ while that of the polygenic *trans* burden is $B\sigma_T^2$; the remaining variance attributable to environmental regulation is $I\sigma_\varepsilon^2$. We estimated these variances using restricted maximum likelihood, as implemented in GCTA (*Yang et al., 2011*), allowing us to quantify the SNP-based *cis* heritability of gene expression as $h_Y^2 = \sigma^2/(\sigma^2 + \sigma_T^2 + \sigma_\varepsilon^2)$. We used DGN whole blood samples ($N = 922$) for maximal power.

### Gradient boosted regression for modeling of evolutionary rate

Selection of informative features among many predictors and incorporation of possibly nonlinear effects into a functional form for evolutionary rate may provide insights into potential causal factors and their relative contributions. The task is a variable selection and model choice problem. Although prediction is not our primary aim in this context, a modeling approach that is robust to overfitting and to redundancy (or multicollinearity) is desired. We therefore modeled dN/dS, based on the human-mouse comparison, using gradient boosted regression trees. The approach combines, in an iterative fashion, otherwise "weak" models or classifiers into a "strong" learner. Using a loss function $L(y, F(x))$ on the evolutionary rate $y$ and the model $F(x)$ built on a vector $x$ of features, the approach seeks to incrementally boost the prediction:

$$M_0(x) = argmin_\alpha \sum_{i=1}^{n} L(y_i, \alpha)$$

$$M_p(x) = M_{p-1}(x) + argmin_h \sum_{i=1}^{n} L(y_i, M_{p-1}(x_i) + h(x_i))$$

 

Here $x_i$ is a vector of features for $y_i$, $n$ is the number of training set observations, $\alpha$ is a constant, $p$ is the iteration index, $M_p$ is the $p$-th model, and $h$ is a base learner (tree) fitted to improve on the model $M_{p-1}$. (The residual $y_i - F(x_i)$ can be interpreted as a negative gradient.)

The models were fitted using the "gbm" R package with a squared error loss function. We utilized a range of values for model parameters. We fitted $N_t$ (100, 500, 1000, 10,000) trees with interaction depth of $D_i(1, 4, 5)$, indicating a model with up to $D_i$-way interactions (which limits the number of nodes in a tree), and a shrinkage parameter of 0.01 (which shrinks the contribution of each tree). We also performed 10-fold cross validation to evaluate the generalizability of the model. Each combination of model parameters provides a "feature importance score", generating a predictor importance spectrum that shows the relative importance of each input feature in predicting evolutionary rate. The feature importance score provides support for how MaxTissue is defined in the proposed tissue-anchored model. In addition, using 100 and 500 iterations, $N_t = 1000$, $D_i = 4$, and shrinkage parameter of 0.01, we trained on 70% of the data to predict the complement set and to obtain a distribution of out-of-sample adjusted $R^2$.

## Gene expression imputation using PrediXcan

PrediXcan is an approach for estimating the genetically determined component of gene expression (*Gamazon et al., 2015a*) using only the germline genetic profile; this estimate is then tested for association with disease risk. Since the disease trait is not likely to modify the germline genetic profile, an observed association comes with a proposed causal direction. Thus, even when directly measured expression is available, its genetic component, though likely to be correlated with the total expression level, provides additional mechanism-relevant information not influenced by disease. Gene expression imputation models were generated using the GTEx reference transcriptome panel in 44 tissues (each with at least 70 samples), as previously described (*Gamazon et al., 2015a*). Each gene expression model (consisting of selected SNPs within 1 MB of the given gene and their additive regulatory effect size on expression) comes with a measure of imputation performance, namely, the 10-fold cross-validation $R^2$ (the square of the correlation between imputed and observed expression) estimated within each tissue (from GTEx or DGN). We note that the best eQTL for a gene also provides an example of an imputation model, though possibly suboptimal in predictive performance. Thus, the practice of testing such an eQTL for its association with a trait is a specific instance, if suboptimal, of the PrediXcan framework. In contrast, our study utilized multi-SNP imputation models for improved performance. The imputation $R^2$ also provides an estimate of the aggregate effect of local genetic variation on gene expression and can be evaluated, in each tissue, for its correlation with evolutionary rate.

## Methodological implications

We assessed several methodological implications on the search for disease-associated genes using both PrediXcan and conventional GWAS. We explored three scenarios.

1. *Extent of local genetic control or cis heritability of gene expression*: *Cis* heritability provides an upper bound on how well local genetic variation may be utilized to impute

gene expression. We calculated the Spearman correlation between the imputation $R^2$ (derived from 10-fold cross validation for each tissue within the PrediXcan framework (*Gamazon et al., 2015a*) and evolutionary rate to evaluate to what extent the genetically determined component of the expression of conserved genes, relative to the complement set, may be reliably imputed. We estimated the reduction in statistical power to detect disease associations for conserved genes (dN/dS < 0.01) relative to fast-evolving genes (dN/dS > 1) in transcriptome-wide association studies, assuming a modestly sized GWAS ($N = 1000$) and using the empirical distribution of $R^2$ for the gene sets in a typical tissue (skeletal muscle, chosen for its sample size). The significance was assessed using the Mann–Whitney $U$ test.

2. *Relevance of cross-tissue imputation models*: We assessed the utility of cross-tissue (versus single-tissue) predictors by evaluating the impact of expression tissue specificity ($\tau$) on gene expression imputation. Genes with $\tau$ close to zero have equal expression levels across all tissues, and in this case, tissue-dependent imputation models are likely to suffer from degraded imputation performance; on the other hand, tissue-specific expression profiles would require tissue-dependent imputation models. We tested for enrichment of broadly expressed genes (defined at multiple thresholds, $\tau < 0.30, 0.40$, and $0.50$) among conserved genes using random sampling from the set of protein-coding genes without replacement ($N = 1000$). For protein-coding genes, $\tau$ has a distribution with a minimum of 0.20 and median of 0.82, indicating high tissue specificity.

3. *Stratified analysis of GWAS data using eQTL information*: Disease-associated loci identified by GWAS have been shown to be enriched for *cis* eQTLs (*Nicolae et al., 2010*; *GTEx Consortium, 2015*). Indeed, incorporation of eQTL information into GWAS analysis has been shown to improve (quite substantially, for some traits and tissues) the false discovery rate. We calculated the Spearman correlation between the absolute magnitude of the largest *cis* genetic effect (within 1 Mb of target gene) and evolutionary rate. A strong positive correlation would suggest, for instance, that conserved genes may be less likely to be detected as eGenes. This observation would have important implications on the task of attribution of gene mechanisms to GWAS loci. We calculated the same correlation for the set of trait-associated SNPs ($p < 5 \times 10^{-8}$) found in the NHGRI GWAS catalog that are in linkage disequilibrium ($r^2 > 0.80$) with a best eQTL using the eQTL's effect size.

## Functional enrichment analysis

We performed functional enrichment analyses using the DAVID Bioinformatics Database (https://david.ncifcrf.gov/) and assuming the human genome as background. We analyzed genes with the lowest 10% of $\tau$ (i.e., high expression breadth) as well as genes that pass a more stringent threshold (i.e., the 100 most widely expressed genes) to find significantly enriched functional annotations and known pathways for tightly regulated genes. At the other end of the distribution, the 100 genes with the most tissue-specific expression profile were evaluated to identify enriched functional annotations. Similarly, we analyzed genes with 1 to 10 interactions to characterize the genes that map to the oldest branch of the
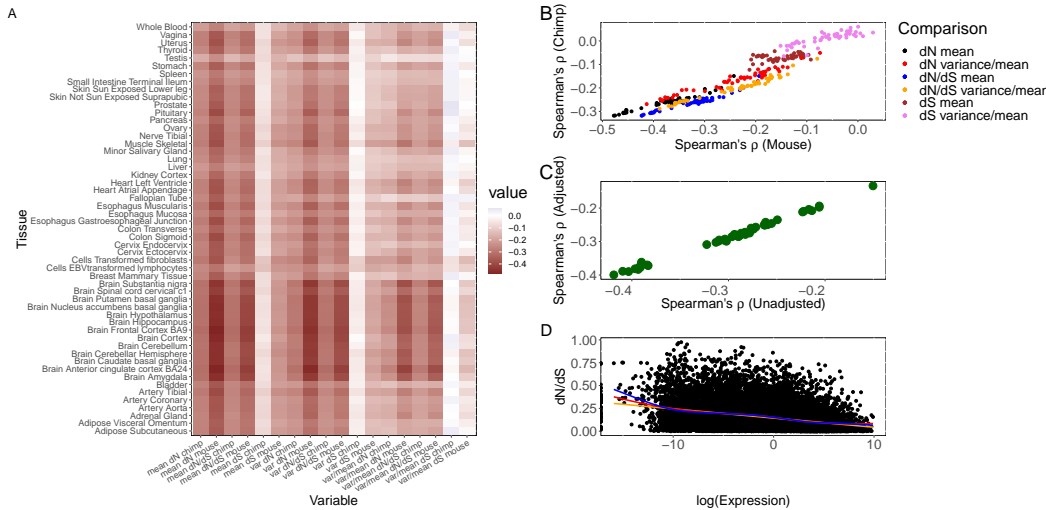

**Figure 1** **Correlation of measures of protein sequence evolution and expression features per tissue.**
(A) Heatmap of (Spearman) correlation of evolutionary rate, nonsynonymous substitution rate, and synonymous substitution rate with expression features. Expression (level or variability) constraints on evolutionary rate and synonymous substitution rate vary widely by tissue and show intertissue correlation. (B) Concordance between the human-mouse and human-chimp comparisons. The comparisons between the two divergence datasets are shown for the correlation of dN, dN/dS, and dS with mean and ratio of variance to mean. (C) Robustness of results from technical confounding. PEER factors were used to quantify hidden and technical confounders. The correlation between expression variance and evolutionary rate is highly correlated across tissues between pre-PEER and post-PEER analysis. (D) Comparison of human-mouse evolutionary rate and $\log_2$ transformed expression level in skeletal muscle. A smooth curve was fitted through the scatter plot with consistent results as the correlation analysis. The orange line denotes fit from least squares regression, red line from LOESS, and blue from Generalized Additive Model.

phylogeny with unexpectedly low connectivity. We also tested the 100 genes at the other end of the node degree distribution to characterize the genes that occupy central network positions.

## RESULTS

### Expression level and variance as tissue dependent correlates of rate of protein sequence evolution

We utilized RNA-seq data derived from 53 human tissues to estimate the mean and variance of expression patterns for each gene (the number of samples for each tissue can be found in Table S1). Throughout the paper, "expression level" refers to the mean expression level. To assess evolutionary conservation, for each orthologous gene we estimated dN/dS rates by comparing human sequences separately with chimpanzee and mouse sequences. We present results from both human-mouse and human-chimp divergence comparisons (Fig. 1), as the human-chimp comparison may be more relevant to the evolution of genes involved in human diseases (discussed later) while the human-mouse comparison is less sensitive to small numbers of mutations.

The rate of nonsynonymous substitution (dN) based on the human-chimp comparison shows a significantly negative correlation (Spearman's $\rho$ ranges from −0.32 to −0.17,

$p < 1.41 \times 10^{-110}$) with mean expression level in all 53 tissues. Whenever we present a range for a test statistic, such as Spearman's $\rho$, for all tissues, the $p$-value presented is always the highest or least significant. The same observation holds (Spearman's $\rho$ ranges from $-0.32$ to $-0.15$, $p < 1.35 \times 10^{-77}$) for evolutionary rate, dN/dS. Stronger correlations with similar trends across tissues are obtained from the human-mouse comparison (Spearman's $\rho$ ranges from $-0.48$ to $-0.24$, $p < 1.37 \times 10^{-235}$ for dN and expression level, and Spearman's $\rho$ ranges from $-0.42$ to $-0.19$, $p < 1.35 \times 10^{-148}$ for dN/dS and expression level).

The uniform analytic pipeline applied to the breadth of tissues (from a large number of the same individuals) enables us to investigate differences between tissues in the correlation between expression in a tissue and support for either purifying selection or positive selection. For each gene in a given tissue, we calculated the mean (to quantify level) and variance or ratio of variance to mean (to quantify inter-individual variability) of gene expression. Notably, the observed expression correlation with sequence divergence varies considerably across tissues and displays tissue clustering (Fig. 1A; human chimp divergence). The brain regions (as previously noted *Duret & Mouchiroud, 2000*) and, interestingly, cervical spinal cord show the largest absolute magnitude of correlation of expression with evolutionary rate while testis, whole blood, and liver are at the other end of the distribution in both divergence comparisons. The clustering for the remaining tissues enables comparisons of the contribution of physiological systems, including reproductive, immune, and gastrointestinal systems, to sequence divergence (Table S2). Notably, although neural tissues tend to cluster together, gastrointestinal systems and reproductive tissues do not. After the neural tissues, the alimentary canal –in particular, the muscularis mucosae of the esophagus, gastro-esophageal junction, and sigmoid colon –has among the largest absolute effects of expression in primary cells on nonsynonymous substitution rate in both divergence datasets.

We examined the tissue distribution of the correlation of expression with silent substitution rate. Selection on synonymous sites has been previously noted in other taxa (*Rocha & Danchin, 2004*; *Larracuente et al., 2008*; *Zhou et al., 2016*). A negative correlation is observed for dS in most tissues (Table S2) though at lower magnitude than for dN despite the strong correlation between dN and dS (Spearman's $\rho = 0.415$, $p < 1 \times 10^{-300}$) based on human-chimp divergence. However, this relationship may merely be due to selective constraints on amino acid changes. Previous studies have indeed attributed the correlation between dN and dS to neighboring effects (*Duret & Mouchiroud, 2000*), and suggested that neighboring sites may influence a site's mutation rate (*Aggarwala & Voight, 2016*). Interestingly, the tissue patterns of the correlation of gene expression with synonymous divergence differ from that with non-synonymous divergence. For example, the highest absolute correlation with dS, based on human-chimp divergence, is in fibroblasts, lymphoblastoid cell lines (LCLs), and skeletal muscle (and not in any of the brain regions); in contrast, the cortex now shows the lowest absolute correlation (Table S2). The tissue dependence of the variation of silent substitution rate with expression becomes more significant when using human-mouse divergence, with aorta, sigmoid colon, and tibial nerve (for example) displaying significantly higher absolute effects than the brain

regions (Fig. 1A). Collectively, these results support the notion that silent sites may be to some degree under selective constraints.

We assessed the contribution of inter-individual transcriptional variability within each tissue, using both the variance and the ratio of variance to mean (given the correlation between expression mean and variance), to evolutionary rate. Expression variance explains similar levels of variation in evolutionary rate as expression mean in all tissues (Fig. 1B), as expected from the correlation between expression mean and variance. Highly significant results on the effect of expression variance in each tissue on nonsynonymous substitution rate and evolutionary rate are obtained with human-mouse divergence (Spearman's $\rho$ ranges from $-0.46$ to $-0.18$, $p < 2.20 \times 10^{-126}$ for dN and expression variance, and Spearman's $\rho$ ranges from $-0.42$ to $-0.15$, $p < 5.20 \times 10^{-90}$ for dN/dS and expression variance). Using the ratio of variance to mean to quantify variability, we continue to observe a significant effect on evolutionary rate in each tissue (Spearman's $\rho$ ranging from $-0.39$ to $-0.08$, $p < 3.49 \times 10^{-26}$).

To evaluate the robustness of our conclusions, we tested the extent to which hidden and unmeasured confounders in gene expression measurements, including any population structure, may bias our analyses. We calculated the Probabilistic Estimation of Expression Residuals (PEER) factors (*Stegle et al., 2012*) and utilized the residual (thereby reducing the impact of technical artefacts, population structure, and other hidden confounders that may be present in the measurement of gene expression) for downstream analysis. We continued to observe a significant correlation between variance (using residual expression after PEER adjustment) and evolutionary rate in each tissue (Fig. 1C), with results between the pre-PEER and the PEER-adjusted analysis being significantly correlated (Spearman's $\rho = 0.997$).

The central nervous system (CNS) tissues, including the brain and spinal cord, consistently show the largest absolute effects of within-tissue expression variability on evolutionary rate using either variance or the ratio of variance to mean (Table S2 and Fig. 1A), but there is also significant variation in observed effects across the CNS tissues. Based on the human-mouse comparison, highly variable genes (defined as the top 1000 in expression variance) in the cortex are under significantly greater purifying selection than highly variable genes in the subcortical regions (Wilcoxon rank sum $p = 1.4 \times 10^{-4}$). On the other hand, highly variable genes in the cervical spinal cord evolve more rapidly than highly variable genes in the brain regions (Wilcoxon rank sum range from $p = 5.17 \times 10^{-9}$ for anterior cingulate cortex BA24 to $p = 0.007$ for substantia nigra). Whole blood and LCLs tend to show lower contribution of expression variance to variation in evolutionary rate than solid tissues, suggesting that evolutionary analysis in these relatively accessible tissues may not accurately reflect the transcriptome-wide contribution to molecular evolution. Although cell type heterogeneity and Epstein-Barr virus transformation may globally alter gene expression profiles in whole blood and LCLs respectively, we note that spleen, a key component of the immune system, also displays among the lowest absolute effects of expression variance on evolutionary rate (Spearman's $\rho = -0.217$, $p = 3.45 \times 10^{-187}$ compared to Spearman's $\rho = -0.419$ for anterior cingulate cortex). Of interest to pharmacogenomics studies, highly variable genes in the liver, a key tissue for

drug metabolism and transport (*Chhibber et al., 2017*), tend to show relatively accelerated evolution (Wilcoxon rank sum $p = 2.33 \times 10^{-15}$ for the comparison of muscle and liver), consistent with the hypothesis that pharmacogenes have evolved as a defense mechanism against the accumulation of harmful xenobiotics.

## A gene's MaxTissue is significantly associated with rate of protein sequence evolution

We find that the tissue ("MaxTissue") in which a gene attains its maximum inter-individual variance ("MaxVariance") –and perhaps also the tissue in which the gene shows the full range of its functional activity –is significantly associated with evolutionary rate (human-mouse comparison, Kruskal–Wallis test $p = 5.55 \times 10^{-284}$). (We note that tissue-specific ($\tau > 0.95$) essential genes demonstrate significantly higher expression variance than tissue-specific non-essential genes (e.g., Mann–Whitney U $p = 1.96 \times 10^{-60}$ in whole blood), supporting the potential value of considering the tissues of high or maximal variance.) The significantly greater variation in evolutionary rate across MaxTissues (than within) suggests a *tissue-anchored model* (see Methods) in which the developmental program or physiological process in which transcription takes place may constrain protein sequence evolution. To illustrate, among 3 selected MaxTissues (cortex, liver, and testis) with significant variance ($p = 5.55 \times 10^{-284}$) in median dN/dS of the genes in a MaxTissue, the 100 genes with the highest expression variance from each MaxTissue are significantly enriched (Benjamini–Hochberg FDR $< 0.10$) for non-overlapping sets of functional annotations (see Methods and Fig. 2A). In cortex, *dendrite*, *cell junction*, and *neuron projections* are among the functional categories implicated ($p = 4.6 \times 10^{-3}$); in liver: *blood microparticle, complement and coagulation cascades*, and *metabolism of xenobiotics by cytochrome P450* ($p = 3.7 \times 10^{-10}$); in testis: *spermatogenesis, chromosome condensation*, and *multicellular organism development* ($p = 5.74 \times 10^{-4}$). The non-overlapping functional categories and the significant difference ($p = 5.55 \times 10^{-284}$) in median evolutionary rate between MaxTissues suggest that variation in the conservation of certain MaxTissue-specific biological and developmental processes significantly contributes to the variation in the rate of protein evolution. Furthermore, permutation analysis that preserves the gene count in the observed MaxTissue configuration finds no random assignment of gene to tissue (RandomTissue; $N = 1000$ datasets) of the entire protein-coding set that generates a (Kruskal–Wallis) statistic as extreme (see Methods). As expected, the genes with the highest expression variance in RandomTissues show no enriched functional categories. The association is not driven by a single tissue or class of tissues. Using the MaxTissue-$\delta$ metric (see Methods) to estimate the variability in evolutionary rate and to rank the MaxTissues, the cortex (0.075), esophagus (0.061), and vagina (0.061) show among the highest values while fibroblasts (0.013), skeletal muscle (0.014), and whole blood (0.015) among the lowest. The other brain regions are spread throughout (e.g., anterior cingulate cortex (0.016), hippocampus (0.017), and caudate basal ganglia (0.051)). Removing the cortex as MaxTissue, the observed association remains significant ($p = 1.04 \times 10^{-284}$).
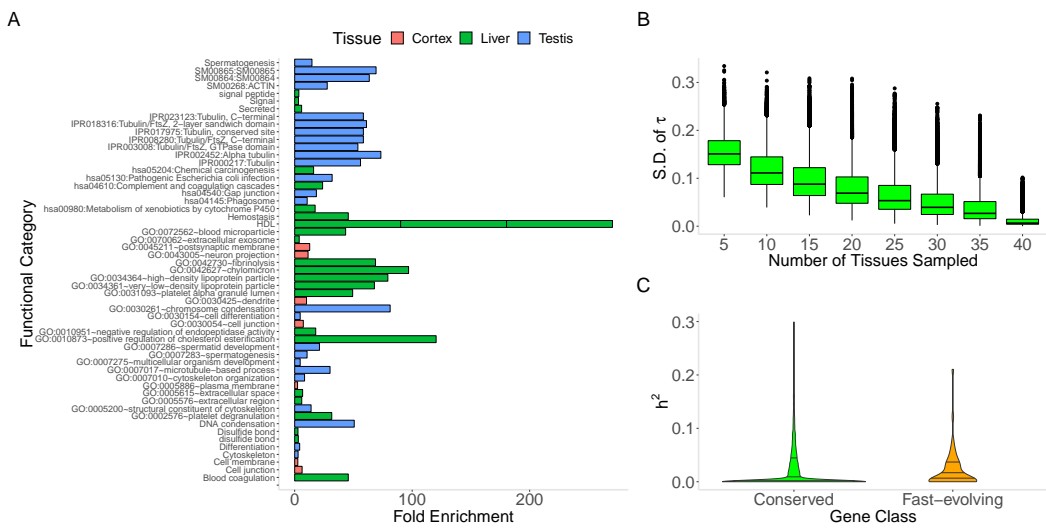

**Figure 2 MaxTissue and the regulatory genome as predictors of evolutionary rate.** (A) Biological and developmental processes implicated by MaxTissues. MaxTissue is significantly associated with evolutionary rate ($p = 5.55 \times 10^{-284}$). The top 100 genes with the highest MaxVariance within 3 chosen MaxTissues—cortex, liver, and testis—are enriched for non-overlapping functional annotations. All functional categories shown here satisfy Benjamini-Hochberg FDR < 0.10 (raw $p$-values range from 0.002 to $6.55 \times 10^{-46}$) and are in the top 25 in each MaxTissue. In cortex, dendrite, cell junction, and neuron projections are among the functional annotations implicated. In liver: blood microparticle, complement and coagulation cascades, and metabolism of xenobiotics by cytochrome P450. In testis: spermatogenesis, chromosome condensation, and multicellular organism development. The non-overlapping enriched functional annotations and the significant difference in evolutionary rate between MaxTissues suggest the importance of MaxTissue-specific biological and developmental processes in constraining evolutionary rate. (B) Estimated tissue specificity of a gene expression trait as a function of number of tissues ($x$-axis). Less comprehensive tissue catalogs can have substantial variability ($y$-axis) in the estimated tissue specificity of gene expression. For a given number of tissues, we randomly selected tissues from the 44 to generate a tissue catalog. We calculated the tissue specificity (tau) of each gene within the tissue catalog. The $y$-axis is the standard deviation of tau across 100 replicates. (C) Regulatory genome and protein sequence divergence. The comparison of local genetic control (measured in DGN whole blood transcriptome) between conserved genes and fast-evolving genes is shown. The first, median, and third quartile are displayed as horizontal lines within the two shaded regions. Conserved genes (defined here as the bottom 25% of the distribution of dN/dS) have significantly lower (Mann–Whitney $U$ $p = 1.2 \times 10^{-17}$) *cis* heritability than fast-evolving genes (defined as the top 25% of the distribution of dN/dS). The median *cis* heritability for all genes is 0.058 (significantly higher, $p = 9.61 \times 10^{-11}$, than for conserved genes). Degree of local genetic control of gene expression is therefore a predictor of evidence for either purifying selection or positive selection.

## A gene's MaxTissue predicts primary affected tissue for developmental disorders associated with the gene

To provide additional insights into how developmental programs within the MaxTissues may constrain protein sequence evolution, as proposed by the tissue-anchored model, we tested the extent to which the MaxTissue for a gene could predict the primary affected tissue for developmental disorders associated with the gene using a large-scale, independently curated resource (see Methods). We found that MaxTissue significantly overlapped ($p = 3.5 \times 10^{-41}$) with the primary affected tissue (Fisher's exact test odds ratio = 4.62), suggesting

a potential mechanism for the tissue-anchored model since developmental disorders are likely to affect fitness and therefore the evolutionary rate of the protein.

In addition to MaxVariance, we tested an alternative cross-tissue feature for its effect on evolutionary rate. We used the tau ($\tau$) statistic to capture the level of similarity of expression across tissues (i.e., expression breadth) in GTEx (*Yanai et al., 2005*; *Kryuchkova-Mostacci & Robinson-Rechavi, 2016*) (see Methods). In the distribution of $\tau$ by gene type (Fig. S1), protein-coding genes display the widest variation; in contrast, lincRNAs show substantial tissue specificity in expression compared to protein-coding genes (Fig. S2). Importantly, using tissue downsampling (see Methods), we find that less comprehensive tissue catalogs than presented here can have substantial variability in the estimated tissue specificity of a gene expression trait (Fig. 2B), with significantly higher variance in $\tau$ among genes with higher $\tau$ (Spearman's $\rho = 0.359$, $p < 1.0 \times 10^{-300}$).

We see a significant positive correlation between dN and $\tau$ (Spearman's $\rho = 0.32$, $p = 1 \times 10^{-300}$) and dN/dS and $\tau$ (Spearman's $\rho = 0.25$, $p = 7.5 \times 10^{-243}$) from the human-mouse comparison, consistent with the notion that intensity of selection on nonsynonymous sites is strongly determined by how broadly expressed the gene is (*Duret & Mouchiroud, 2000*).

We conjectured that the difference in tissue specificity between non-coding and protein-coding genes may indicate functionally relevant differences between regulatory genes and their targets and provide insights into how expression breadth may constrain evolutionary rate. Indeed, functional enrichment analysis of the lowest 10% of $\tau$ scores (i.e., genes with the broadest expression profile) implicates functional annotations relating to transcription (Benjamini–Hochberg adjusted $p = 6.19 \times 10^{-36}$) and mRNA processing (Benjamini–Hochberg adjusted $p = 1.57 \times 10^{-34}$) (*Huang, Sherman & Lempicki, 2009*). More stringently, the 100 genes with the greatest expression breadth are enriched in *Ubl conjugation pathway* ($n = 13$, Benjamini–Hochberg adjusted $p = 3.7 \times 10^{-3}$), a highly conserved eukaryotic gene regulatory mechanism that frequently promotes protein-protein interactions (*Hochstrasser, 2009*). Thus, a possible explanation for the importance of expression breadth in constraining evolutionary rate is the conservation of genes involved in multi-tissue gene regulation. In contrast, the 100 genes with the most tissue-specific expression profile are enriched for secreted proteins ($n = 31$, Benjamini–Hochberg adjusted $p = 8.5 \times 10^{-9}$) with gene products, for instance, located in extracellular region or involved in signaling.

Comparing the two cross-tissue features, we find that $\tau$ explains only 0.88% ($p = 6.0 \times 10^{-40}$) of the variability in MaxVariance. Using EPRCA, we find that evolutionary rate and MaxVariance are significantly correlated after controlling for expression breadth (Spearman's $\rho = -0.151$, permutation $p < 0.001$). These results suggest that expression breadth and MaxVariance, as predictors of between-tissue effects, may independently constrain evolutionary rate.

### Joint feature analysis of rate of protein sequence evolution (dN/dS)

Besides MaxTissue and MaxVariance (above), we identified additional correlates of dN/dS. Heritability provides a measure of the potential of a trait to respond to selection (*Roff, 2000*). Gene expression level as a quantitative trait has a heritable component. For many

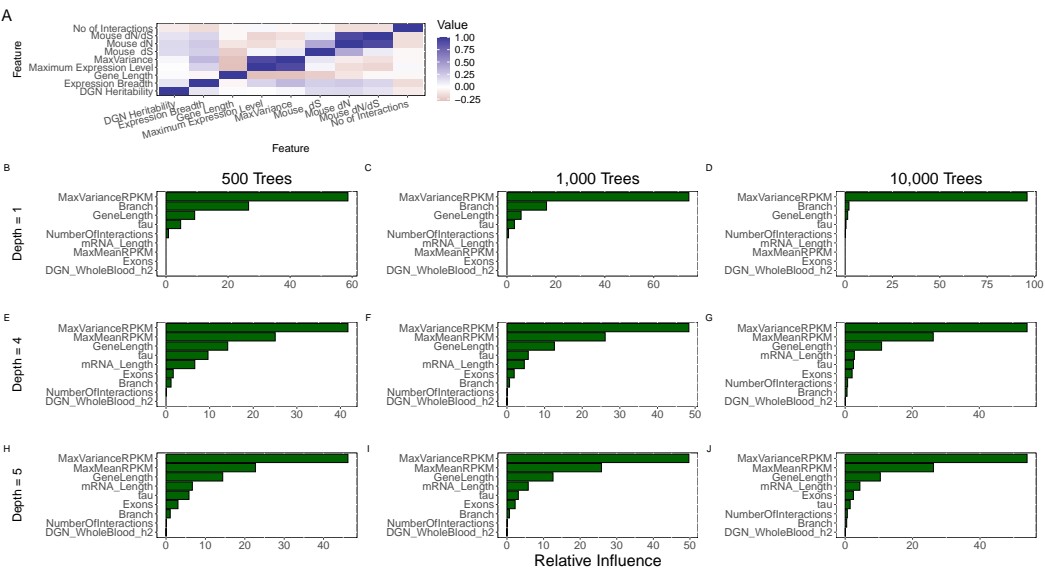

**Figure 3** **Joint analysis of possible determinants of evolutionary rate.** (A) The pairwise (Spearman) correlation values for all features. Expression breadth is given by the tau statistic with smaller tau indicating wider expression breadth. Pairwise correlation was restricted to protein-coding genes (even though some correlations could be calculated with non-coding genes) because of our focus on protein evolutionary rate. (B–J) Comparison of feature importance scores. MaxVariance shows the highest feature importance score using a range of model parameters for the number of trees (500; 1,000; 10,000) and the interaction depth (1, 4, 5) in gradient boosting. In particular, MaxVariance shows a stronger influence on predicting evolutionary rate than the other cross-tissue features: maximum expression level and expression breadth.

genes, expression level can be predicted, to a degree measured by heritability and in a tissue-dependent manner, based on genetic polymorphism data (*Gamazon et al., 2015a*). We find a significant relationship between the *cis* (common variant) heritability of gene expression and dN/dS (Supplementary Information and Fig. 2C). Conserved genes tend to have lower *cis* heritability than other genes (Mann–Whitney U $p = 1.2 \times 10^{-17}$). Although gene age and PPI node degree have been previously investigated (and although the method for gene age estimation may generate methodological artifacts (*Moyers & Zhang, 2017*)), we report their relationship with the unique expression data analyzed here (controlling for potential confounders) and find these variables to be significant associated with dN/dS (Supplementary Information).

The features we have examined are mutually correlated (Fig. 3A), posing a challenge to the search for causal factors. A comparison of the generalized additive models based on the human-mouse comparison using AIC (which is asymptotically equivalent to leave-one-out cross-validation for ordinary linear regression (*Stone, 1977*)), is shown in Table S4. Notably, tissue expression breadth is the model with the best univariate fit.

We then implemented an approach that performs variable selection and model choice (given the number of correlated predictors) and incorporates potentially nonlinear effects (given the univariate observation on the presence of such effects). We utilized gradient boosted regression trees to model evolutionary rate (see Methods), based on human-mouse

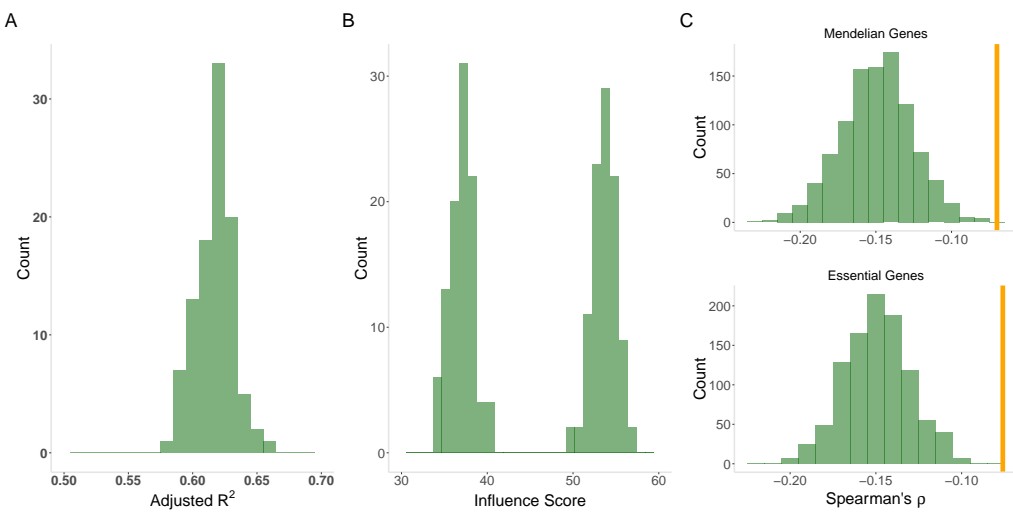

**Figure 4  Model of evolutionary rate.** (A) Variance explained by model. The joint model from gradient boosting explains an average of 62% of the variability in evolutionary rate from out-of-sample prediction. The distribution is from 100 cross validation analyses. (B) Variability versus level. MaxVariance (right histogram) consistently shows a higher relative importance than maximum expression level (left histogram) in determining evolutionary rate using 100 replicates from gradient boosting. We therefore used MaxVariance to define MaxTissue, a key element of the tissue-anchored model of evolutionary rate. (C) Interaction of MaxVariance and function. Mendelian disease genes and essential genes show substantially lower MaxVariance effect (empirical $p < 0.001$), in terms of variance explained, on evolutionary rate than the full set of proteins, indicating that MaxVariance and gene function may interact to constrain evolutionary rate.

divergence, jointly analyzing all the features. Notably, we find that MaxVariance, among all features, has the highest feature importance score using a range of model parameters for the number of trees and the number of interactions (see Methods) (Fig. 3B), providing further support to the proper centrality of this feature in our tissue-anchored model of evolutionary rate. In particular, MaxVariance has a stronger influence on predicting evolutionary rate than maximum expression level. We find that the combined model explains an average (and median) of 62% of the variability in evolutionary rate (Fig. 4A) based on out-of-sample prediction (see Methods), with MaxVariance consistently having a higher relative importance than maximum expression level (Fig. 4B). The relative importance of MaxVariance in the gradient boosted model is consistent with the single-model finding that a significant association with evolutionary rate characterizes the MaxTissue configuration and is not observed for a random tissue reconfiguration (permutation $p < 0.001$; see Methods).

   We also considered specific gene sets to determine to what extent the observed relationships between evolutionary rate and expression features apply (Supplementary Information). Mendelian disease genes and essential genes display substantially lower MaxVariance effect, in terms of variability explained, on evolutionary rate than randomly drawn sets ($N = 1,000$) of proteins, indicating that MaxVariance and gene function may interact to influence evolutionary rate (Supplementary Information and Fig. 4C). As in the

# PeerJ

full set of protein-coding genes, when restricted to Mendelian disease genes for all model assumptions examined, gradient boosted modeling analyses indicate that MaxVariance continues to have a higher relative importance than maximum expression level.

## Methodological implications on genomic studies of disease

Our collective findings have important methodological implications for the search for disease-associated genes and the study of gene function. We explore three specific applications (using the human-chimp comparison for illustration).

The genetically determined component of gene expression in a tissue (see Methods) is typically tested for association with disease in transcriptome-wide association studies (e.g., using PrediXcan (*Gamazon et al., 2015a*)). Our study would suggest that imputation of genetically determined expression for conserved genes using genetic variants within the *cis* region of a gene (1 Mb) should be less accurate than for fast-evolving genes. Genes with less accurate expression imputation would suffer from lower statistical power in an association analysis. We note that even if directly measured gene expression data are available, estimation and inference on the genetically determined expression should yield additional information on underlying disease mechanisms.

Our hypothesis is supported by the significant positive correlation across genes between the out-of-sample imputation $R^2$ for gene expression (see Methods) and dN/dS (Spearman's $\rho = 0.09$, $p = 8.6 \times 10^{-14}$ for adipose subcutaneous; Spearman's $\rho = 0.057$, $p = 2.6 \times 10^{-6}$ for skeletal muscle; and Table S6 for the remaining tissues). For most tissues, imputation performance is significantly lower for conserved genes than for fast-evolving genes (Fig. 5A). The differential performance has important implications for mapping disease-associated genes. Indeed, in a tissue (skeletal muscle, chosen for its sample size) with significant differential imputation performance (Bonferroni-adjusted $p < 0.05$ for number of tissues tested) and a modestly sized genome-wide association study ($N = 1000$), we estimate that conserved genes would have significantly lower statistical power than fast-evolving genes (median power 0.16 versus 0.41, Mann–Whitney U test $p = 2.2 \times 10^{-6}$; see Methods). Furthermore, a smaller proportion of conserved genes (0.32 versus 0.41) would have at least 80% power (Fig. 5B). Finally, the significant correlation with out-of-sample imputation $R^2$ confirms the significant positive correlation between evolutionary rate and *cis* heritability, for which estimation, using linear mixed models, depends on a rather strong assumption of polygenicity of gene expression.

A second application involves the utility of multi-tissue imputation models. We find that genes involved in multi-tissue gene regulation are notable for the relative strength of purifying selection acting on them, contributing to the strong correlation between expression breadth and evolutionary rate. Our study would suggest that imputation of gene expression for conserved genes could be substantially improved using multi-tissue (versus tissue-specific) genetic predictors. Consider the hypothetical case where a gene is broadly expressed and, indeed, has equal expression across all tissues (i.e., $\tau = 0$). Clearly, an imputation model that varies by tissue would be inconsistent with the gene's true expression profile and, thus, have suboptimal imputation performance. On the other
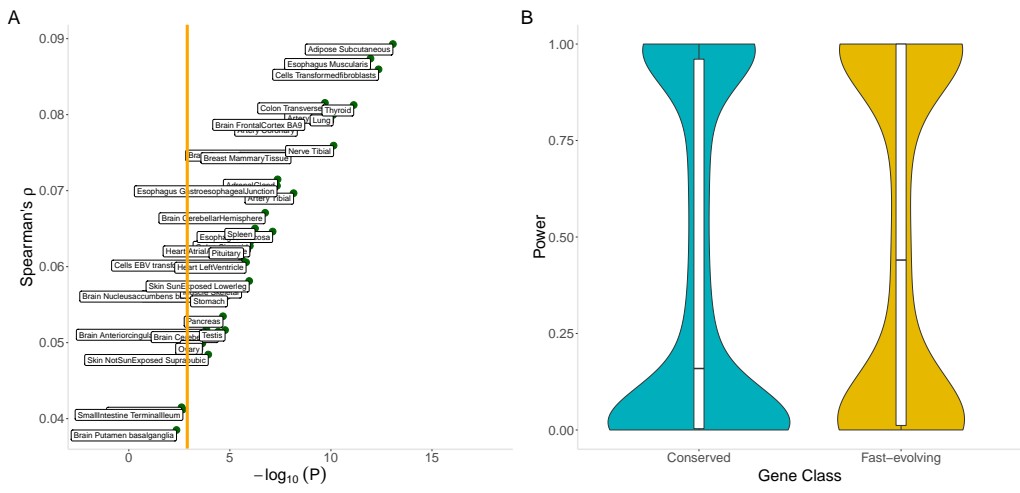

**Figure 5 Correlation between evolutionary rate and out-of-sample imputation $R^2$ for gene expression.** (A) The out-of-sample $R^2$ assumes local predictors and is derived from 10-fold cross validation within the PrediXcan framework applied to GTEx data. For a wide range of tissues, out-of-sample imputation accuracy is significantly correlated with evolutionary rate. In particular, this suggests that transcriptome-wide association analyses using local predictors in a range of tissues have significantly less power to detect disease associations for conserved genes than for fast-evolving genes. Orange line indicates Bonferroni threshold for the number of tissues tested. (B) Conserved genes have significantly lower statistical power than fast-evolving genes (median power 0.16 versus 0.41, Mann–Whitney $U$ test $p = 2.2 \times 10^{-6}$). Furthermore, a smaller proportion of conserved genes (0.32 versus 0.41) would have at least 80% power. Here skeletal muscle was used.

hand, a gene with highly tissue-specific expression profile (e.g., $\tau = 1$) would require tissue-dependent models for optimal imputation.

Consistent with this hypothesis, among the most highly conserved genes (human-chimp dN/dS $< 0.01$, $N = 2,750$), a significant enrichment ($p < 0.001$) for broadly expressed genes can be found (see Methods). In general, 4.41% of the variance in evolutionary rate is explained by expression breadth. These genes that are subject to the strongest purifying selection show a significant 15% decrease in tissue specificity ($\tau$) (Wilcoxon rank sum $p = 9.57 \times 10^{-78}$) relative to the remaining protein-coding genes.

For a final application, we investigated whether our findings would also yield methodological insights into traditional single-variant analyses of complex traits. Since trait-associated SNPs identified by GWAS have been shown to be enriched for *cis* eQTLs, we asked where the target genes of these regulatory variants lie in the overall distribution of evolutionary rate. We calculated the Spearman correlation between the best *cis* eQTL effect (within 1 Mb of target gene) in absolute value and evolutionary rate. We find a significant positive correlation (Spearman's $\rho = 0.204$ between absolute *cis* effect size and dN/dS, $p = 1.3 \times 10^{-133}$), suggesting that conserved genes may be less likely to be detected as eGenes. Furthermore, using reproducible trait-associated variants ($p < 5 \times 10^{-8}$) curated in the NHGRI GWAS catalog that are in linkage disequilibrium ($r^2 > 0.80$) with a best eQTL, this correlation becomes even greater (Spearman's $\rho = 0.36$, $p = 6 \times 10^{-32}$), suggesting that conserved genes may be under-represented as target genes of variants identified by

genome-wide association studies. This finding raises the possibility that attribution of gene mechanisms to GWAS loci using only *cis* eQTLs has underestimated the influence of this class of genes.

## DISCUSSION

The GTEx resource consisting of RNA sequencing data collected in multiple individuals across 53 different tissues permitted a comprehensive analysis of protein sequence evolution for genes in diverse physiological systems. We examined a variety of expression-based models of evolutionary rate in a comprehensive multi-tissue framework, evaluated the relevance of the framework to characterizing human disease-associated genes, and explored a number of methodological implications for transcriptome-wide association studies (*Gamazon et al., 2015b*; *Gusev et al., 2016*). We also present a tissue-anchored model, with MaxTissue a significant tissue configuration for evolutionary rate, and a combined feature set that explains a large proportion of the variation.

The greatest effects of expression level, in terms of variance explained, on evolutionary rate are observed for the CNS, including the brain, consistent with an earlier report (*Duret & Mouchiroud, 2000*; *Gu & Su, 2007*; *Tuller, Kupiec & Ruppin, 2008*; *Kryuchkova-Mostacci & Robinson-Rechavi, 2015*), and cervical spinal cord. However, we also find a remarkably similar level of contribution from expression variance in the CNS tissues, which remains significant after controlling for the mean level. The sampling of diverse CNS tissues enabled us to observe significant variation in expression constraints on sequence divergence. Notably, highly variable genes in the cortex are under significantly greater purifying selection than highly variable genes in the subcortical regions, illustrating differences in evolutionary conservation for this class of genes even among related neural tissues. The higher absolute effect of expression level or variance on evolutionary rate observed in the CNS system than in other systems may be in part attributable to increased selection to prevent toxicity from protein misfolding and aggregation of highly expressed genes.

A uniform analytic pipeline applied to the diversity of tissues provides a unique opportunity to estimate the contribution of transcriptional variance in each tissue to evolutionary rate. We tested both variance and ratio of variance to mean and observe similar tissue-dependent patterns. Several mechanistic hypotheses, for the role of expression level in constraining evolutionary rate, have been proposed (*Akashi, 2003*; *Rocha & Danchin, 2004*; *Lemos, Meiklejohn & Hartl, 2004*; *Drummond et al., 2005*). On the other hand, expression variance captures the range of expression level over which a gene functions in a given tissue and thus potentially reflects a temporal component to expression or a cellular or developmental process. Instead of maintaining a steady-state value near the mean, some conserved genes may express at low levels and then quickly express to high levels when activated. Rapid cycling of expression of such genes in a tissue may also therefore play a role for the protein folding accounts previously proposed for expression level. In contrast with these within-tissue predictors of evolutionary rate, the association of tissue (MaxTissue) with maximum expression variance with evolutionary rate, which our empirical analysis shows to be a significant tissue configuration, highlights the importance

of cross-tissue predictors and the developmental or physiological component of the variation in evolutionary rate.

Here, we measure expression breadth in tissues derived primarily from adults (*GTEx Consortium, 2015*), which may underestimate the true expression breadth of genes that have a strong temporal component to their expression. Importantly, tissue downsampling shows that expression breadth for genes expressed in a large number of tissues is robust to tissue sampling but, for genes expressed in a limited set of tissues, is highly sensitive to the choice of tissues. This vulnerability of tissue-specific genes to mismeasurement underscores the utility of a large-scale resource in assessing the importance of tissue breadth, as previous studies of protein sequence evolution have examined only a small number of tissues. This has further implications for studies of younger genes (e.g., primate-specific), which tend to have more tissue-specific expression. In a comparison of cross-tissue predictors (versus within-tissue expression features), expression breadth accounts for only a small proportion (8.9%) of the variability in MaxVariance, which may therefore represent independent constraint and, indeed, shows the greatest contribution to predicting evolutionary rate among all features.

We find, using whole blood transcriptome data, that *cis* heritability of gene expression is significantly positively correlated with nonsynonymous substitution rate as well as with evolutionary rate. Genes with low *cis* heritability of expression tend to be conserved. However, the linear mixed model approach to SNP-based heritability estimation assumes a polygenic *cis* architecture of gene expression; this approach may therefore perform poorly in the presence of large-effect regulatory variants. Nevertheless, the out-of-sample imputation $R^2$ (which provides an estimate of the *cis* heritability and should be more robust to model misspecification) from the gene expression (PrediXcan) model also varies with evolutionary rate. One explanation for the observed relationship between heritability and evolutionary rate is that conserved genes have less standing genetic variation in *cis* in a population, and therefore, lower heritability. However, the estimated heritability does not include the effects of more complex forms of genetic variation or of rare regulatory variation. The tissue variation in the correlation of the imputation $R^2$ (as an estimate of heritability) with evolutionary rate suggests that the implied effect of the regulatory genome on protein sequence evolution may in part arise from genetic control of tissue-specific biological processes. For example, genes that underlie specific developmental programs may evolve significantly more slowly than those for other developmental programs, contributing to variation in evolutionary rate.

Notably, the correlation of evolutionary rate with gene expression heritability and with MaxTissue are not predicted by the translational robustness hypothesis (*Drummond et al., 2005*), which proposes that highly expressed genes are conserved due to selection against protein misfolding caused by nonoptimal amino acids. Importantly, the translational robustness hypothesis leaves unanswered the question of why the correlation between expression and evolutionary rate should, as reported here, significantly vary by tissue. Our proposed tissue-anchored model highlights the centrality of tissue-specific biological processes and indicates that a specific tissue configuration, i.e., the one that maps a gene to its tissue of maximum variance (thus, the tissue which perhaps most accurately reflects

the gene's range of functional activity), may be unique for its constraint on evolutionary rate. The observed constraint of the regulatory genome (with its potential tissue specificity) on evolutionary rate and the evidence for the tissue-anchored model would suggest the importance of the developmental and physiological processes in which transcription takes place (thus determining gene function in a context-specific way) as a primary driver of protein sequence evolution in humans. Notably, using a reference UK-wide genomic resource, we find that the MaxTissue for a gene significantly predicts the primary affected tissue for developmental disorders associated with the gene, lending substantial empirical support to the tissue-anchored model since developmental disorders are likely to affect fitness and thus evolutionary rate.

We explored several methodological applications of our evolutionary analysis to genomic studies of disease. Gene expression imputation (*Gamazon et al., 2015a*; *Wang et al., 2016*) is a powerful approach to mapping disease-associated genes that is now becoming more widely used (*Hoffman et al., 2017*; *Son et al., 2017*; *Xu et al., 2017*; *Xu et al., 2018*; *Zeng, Wang & Huang, 2017*; *Gottlieb et al., 2017*; *Li et al., 2018*; *Sanchez-Roige et al., 2018*; *Lamontagne et al., 2018*). In this framework, the imputed genetically determined expression in a tissue is tested for its contribution to disease phenotype. An observed correlation with the genetically determined component (unlike the directly measured expression level) proposes a causal direction of effect, as the disease trait is not likely to alter the germline genetic profile. (Of course, definitive identification of causal genes may ultimately require perturbation experiments.) Thus, estimating the genetically determined component of gene expression extends differential expression analysis of the directly measured gene expression. As predicted by our framework, the expression of conserved genes is not as accurately imputed, using local genetic variation, as the expression of fast-evolving genes. Thus, transcriptome-wide association studies, such as implemented in PrediXcan (*Gamazon et al., 2015a*), in search of disease-associated genes may be substantially enhanced by incorporating the possible determinants of evolutionary rate into the analysis. Since Mendelian disease genes are enriched for conserved genes, the lower imputation quality for these genes implies that their reported effects on complex traits from local genetic variation may be severely underestimated despite the substantial comorbidity associations between Mendelian and complex disorders that have been uncovered (*Blair et al., 2013*).

A major methodological implication of our study is the importance of building evolutionary-rate-aware models of genetically determined expression for increased statistical power. Fast-evolving genes tend to be highly tissue-specific, suggesting the importance of tissue-specific imputation models for these genes. On the other hand, genes under strong purifying selection, including essential genes, tend to be broadly expressed, suggesting the importance of multi-tissue imputation models that explicitly utilize their tissue-shared expression profile. Current models, which are indifferent to evolutionary rate and its contributing features, are thus likely to be underpowered for a class of genes.

Here we provide a comprehensive transcriptome-based analysis of the factors that may constrain protein sequence evolution. We highlight the importance of cross-tissue predictors, whose properties can be more accurately characterized using a more comprehensive tissue collection than previously available and a uniform analysis pipeline.

Importantly, we show, using gradient boosted regression, that the feature set explains 62% of the variability in evolutionary rate and provides additional support to the tissue-anchored model. The tissue-anchored model reinforces the notion that rather than a single mechanism (gene expression) being significant for protein evolution, variation in the conservation of certain developmental programs in which transcriptional dynamics unfolds in space and time may be a primary driver of evolutionary rate. Mechanistically, the regulatory genome may constrain evolutionary rate through its overall effect on the phenome (*Nicolae et al., 2010*; *GTEx Consortium, 2015*). Contrary to studies suggesting that evolutionary rate is primarily driven by constraints on protein or mRNA folding (*Drummond et al., 2005*; *Yang et al., 2012*), Mendelian disease and essential genes show significantly lower absolute effect of expression on evolutionary rate than random genes. This observation, coupled with the tissue-anchored model, supports the greater role of gene function than of protection from protein misfolding, in the evolution of these genes.

Our study used species divergence to measure protein evolution and selection, but we have not utilized the polymorphism data within GTEx to examine signatures of more recent selection (*Voight et al., 2006*; *Field et al., 2016*). It would be interesting to test whether our conclusions hold at shorter human evolutionary time-scales. Furthermore, the use of additional species (given the availability of dozens of mammalian genomes now) may improve the evolutionary analyses presented here. We also examined mRNA expression, which may not reflect protein-specific regulatory mechanisms. Incorporation of protein expression, translational efficiency, and codon bias will likely improve our model of protein evolution and provide additional mechanistic insights into disease biology.

## CONCLUSIONS

We evaluated several expression-based models of protein evolutionary rate in a multi-tissue framework, assessed their relevance to characterizing human disease-associated genes, and explored methodological implications for transcriptome-wide association studies. We propose a tissue-anchored model for protein evolutionary rate and a combined feature set that explains a large proportion of the variation. The tissue-anchored model provides a transcriptome-based approach to predicting the primary affected tissue of developmental disorders, as we confirmed using a large-scale and independently curated resource, suggesting a potential mechanism since developmental disorders are likely to affect fitness and thus the evolutionary rate of the protein.

## ACKNOWLEDGEMENTS

We would like to thank Jibril Hirbo for his editorial feedback. E.R.G. is grateful to the President and Fellows of Clare Hall, University of Cambridge for providing a stimulating intellectual home during his Visiting Fellowship.

### Funding

This work was supported by the National Human Genome Research Institute of the National Institutes of Health (R35HG010718). The funders had no role in study design, data collection and analysis, decision to publish, or preparation of the manuscript.

### Grant Disclosures

The following grant information was disclosed by the authors:
National Human Genome Research Institute of the National Institutes of Health: R35HG010718.

### Competing Interests

Eric R. Gamazon receives an honorarium from the journal Circulation Research of the American Heart Association, as a member of the Editorial Board. He performed consulting on pharmacogenetic analysis with the City of Hope / Beckman Research Institute.

### Author Contributions

- Patrick Evans and Eric R. Gamazon conceived and designed the experiments, performed the experiments, analyzed the data, prepared figures and/or tables, authored or reviewed drafts of the paper, and approved the final draft.
- Nancy J. Cox analyzed the data, authored or reviewed drafts of the paper, and approved the final draft.

### Data Availability

The protected data for the GTEx project (for example, genotype data) are available on request: dbGaP accession number phs000424.v6.p1. Access to GTEx V8 protected data requires an approved dbGaP application: https://gtexportal.org/home/protectedDataAccess.

Processed GTEx data (for example, gene expression and eQTLs) are available on the GTEx portal: https://gtexportal.org/home/datasets.

The remaining data are in the Supplementary File and at GitHub: https://github.com/gamazonlab/EvolutionaryRate_RegulatoryGenome.

### Supplemental Information

Supplemental information for this article can be found online at http://dx.doi.org/10.7717/peerj.9554#supplemental-information.

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
