# Peer review of "The regulatory genome constrains protein sequence evolution: implications for the search for disease-associated genes"

_PeerJ, doi:10.7717/peerj.9554_

## Round 0.1 · original submission · Minor Revisions

The reviewers have appreciated the methodology and approach as well as attention to an important topic, but have raised minor comments, mainly on clarity of results and availability of data and code, that should be addressed before the paper is suitable for publication.

·

Basic reporting

Really solid supplementary data reporting. Only a few notes: Supp Table 3 appears to be a “.tsv” not “.xlsx” (my computer had trouble opening it so just double check that it’s format is corrected). I also feel that it would be a pretty hefty table but a similar .tsv file for each gene that includes the mean, median, and variance in expression for all 53 tissues would be really nice to have too.

Should provide R code used in analysis or a link to it.

Missing some important references with regard to the tissue-specificity in the evolutionary rate/expression relationship. Lots of folks have looked at / questioned this issue of tissue specificity and think the authors could better characterize what is known up to this point. See, for instance:

https://journals.plos.org/plosone/article?id=10.1371/journal.pone.0131673
https://genomebiology.biomedcentral.com/articles/10.1186/gb-2008-9-9-r142
https://link.springer.com/article/10.1186/1471-2148-10-241
https://www.pnas.org/content/104/8/2779

Experimental design

Overall a well defined and well researched paper. Methods section is very well written and it is clear that the authors have a firm understanding of the statistics and statistical issues.

Validity of the findings

Well written throughout and all conclusions are within the bounds of what the authors specifically show. Great job overall.

Additional comments

Summary:
In this manuscript, Evans et al. leverage a large resource of tissue-specific gene expression to investigate the determinants of evolutionary rate variation within humans. Overall, I found the research to be interesting and well-performed, and by and large found the manuscript to be well written. Most of my below points are fairly minor in scope but I’ll nevertheless dump them here for the authors to do with as they see fit.

First the figures: I found these to be a little bit messy overall. I can’t say that any of this stuff *needs* to be addressed prior to publication but in my opinion it could really be cleaned up. Some labels are really tiny/bordering on illegible (legend in 1B, axes and ticks in 2C). Some of the images look a bit fuzzy too, but this could just be a formatting thing at this stage. Figures follow a range of styles, even directly next to each other. See for instance the labels in 2B vs 2C or 4A vs 4B (in terms of whether to bold axes labels, and 4C actually uses a mixture of both). In 3B the “tree depth label” is just kind of hanging there awkwardly. In 4B there should be a legend saying what the two colors are or at least mention these in the caption. Finally, an opinion of mine is that I think the manuscript could be strengthened with some example data in Fig. 1. The data presented throughout is kind of abstracted from the basic level of essentially comparing scatterplots/correlations between variables. I feel that two example scatterplots for like the strongest and weakest rho tissues could help to show what these relationships actually look like rather than just always thinking of them as rho’s and p-values. Clearly can’t/shouldn’t do it for everything but examples always help with interpretation.

Line 47 – “A second engine of protein evolution, fixation of new mutations depends upon effect on fitness, genetic drift, and population size.” – probably better stated as something along the lines of “depends upon the fitness effect of the relevant mutations and the balance between selection and genetic drift, which is in part mediated by effective population size.”

Line 52 – Authors note that the assumption of dS being fully neutral is likely violated, think I would just draw a bit more attention to it with some references. But state that the reality is that in spite of the over-simplification of dN/dS it’s still one of the most commonly used and “acceptable” frameworks. I also feel like this paragraph is better suited to the discussion section, especially since dN/dS is so commonly used and the authors aren’t actually performing their own evolutionary analyses and rather are using the values from biomart. At very least think this section of the intro could be cut down as we’re in nitty-gritty protein evolution methods territory quickly before really getting to the actual problem the authors want to address.

Line 57 – Looks like the period is red.

Line 60 – Feel like there should be a lead in / discussion more about single-celled organisms. A lot of early protein evolution work was done on single-celled yeasts and bacteria. But the presence of tissues with unique expression profiles makes studying similar processes difficult in multi-cellular eukaryotes, a discussion of which would lead well in to this paragraph.

Line 68 – I really like this paragraph. But also think it’s another opportunity to better frame the research by questioning what “expression level” means for multi-cellular/multi-tissue organisms with long-lifespans and developmental processes (in contrast to, again single-celled organisms). As is obvious by this point, this is what I see as the big outstanding question that this paper really seems to be addressing without ever really explicitly stating it.

Line 343 – The beginning of this results section is really jarring and feels tacked on by a previous review or something. The first line there just kind of jumps out and I think the readability of the manuscript would be much improved with a brief re-iteration of what the major points of the analysis are here. My guess is that ~1% of the readership of the paper will likely read through the methods so throughout the results section I’d encourage the authors to write as if that were the case by re-iterating basic steps of the analysis at a high level to help readers follow the logic. i.e.:
“We compiled RNA-seq samples for 53 human tissues to estimate the range of expression patterns experienced for each gene (the number of samples for each tissue can be found in Supplementary Table 1). To assess evolutionary conservation, for each orthologous gene we estimated dN/dS rates by comparing human sequences separately with chimpanzee and mouse sequences. The human-chimp comparison may be more…”

Line 348 – “Rate…”=> “The rate…”

Line 349 (and 351, 352, 353) – It’s a bit unclear what the p-value there is, is it a meta-p-value for all 53 tissues? Or the p-value for the strongest or weakest correlation? This appears throughout when the authors are presenting a range of values and I think an explanation should just appear in that first one.

Line 350 – “with (mean? median?) expression level in all 53…”

Line 535 – This is a really nice section that you often wouldn’t find in similar work.

Reviewer 2 ·

Basic reporting

no comment.

Experimental design

no comment.

Validity of the findings

no comment.

Additional comments

The authors performed a GTEx transcriptome analysis across multiple tissues to assess the effect of the transcriptome in protein sequence evolution. They analysed several expression-based models of protein evolution across tissues, examined the potential of this approach to be used to identify disease-related genes, and discussed the methodological implications for transcriptome-wide association studies of disease. The authors propose a tissue-based model of protein sequence evolution, where the correlation between gene expression levels and evolutionary rate varies significantly by tissue.

This study delivers a very sound approach to better assess and quantify the effect of gene expression on protein sequence evolution, with implications for the prediction of disease-associated genes. I just recommend clarifying the below-mentioned issues before publication.


As a general comment, throughout the manuscript, you use two types of divergence without any explanation of why you use one or the other (except for disease-associated genes). For most of the analyses, you use the human-mouse divergence data, but it’s unclear why.

Minor comments:
Introduction:
Lines 47-48: unclear sentence, I propose to rephrase it as: “The fixation of new mutations depends on its fitness effect, (…)”.

Lines 54-56: This sentence needs to be clarified. Using the dN/dS ratio is a conservative approach because it only provides an average of the substitution rate across a certain number of sites. Since most of the amino-acid mutations are either neutral or deleterious, dN/dS will generally be lower than 1, hence potentially overlooking positions under positive selection.

Methods:
Line 93: Were the inter-individual variability analyses performed on human individuals? If so, how many? This is not clear in the text.

Line 99: define “LCLs” here, you have it only in the results section.

Line 107: define “PEER” here, you have it only in the results section.

Line 208: by correlation in gene expression, you mean genes that belong to the same gene network?

Lines 221-223: if I understood correctly, you mean here that you use the GLS to account for the non-independence of genes in a gene network, like in a phylogenetic statistical approach, is that right? If so, it needs to be better clarified. It may not be straightforward for the reader to understand this.

Lines 284-285: this sentence is unclear to me. What do you mean by modifying the germline genetic profile? I thought that the software only provided a prediction for the association with a disease trait.

Results
Lines 348-354: If I understood correctly, you estimated the mean dN and dN/dS across genes in a tissue. Why not use the median here? Were these estimates corrected for the number of genes expressed in each tissue? Just for a sanity check, you could perform the same analyses by randomly downsampling the same number of genes in each tissue (say 100 times) and then estimate the mean value dN and dN/dS of the bootstrap replicates.

Lines 376-378: This sentence could be simplified. You could say something like, “The tissue patterns of the correlation of gene expression with synonymous divergence differ from that with non-synonymous divergence”.

Lines 381-384: Not very clear. Do you mean that the correlation between expression levels with the synonymous substitution rate for each tissue is stronger when using the human-mouse divergence?

Line 397: population structure between which populations?

Line 399: what do you mean by the impact of batch effects?

Lines 425-428: I don’t understand how the Kruskal-Wallis test was performed here. Which groups are you comparing? And what do you mean by associated with the evolutionary rate? Do “MaxTissues” have genes with higher evolutionary rates in general?

Line 434: I would remove “namely” from the brackets to make it clearer.

Lines 434-436: confusing, do you mean significant variance in dN/dS between genes in a MaxTissue? 100 of what? Do you mean the 100 genes with the highest expression variance?

Line 493: I would replace “that capture” with “of” to simplify the sentence.

Lines 497-500: I would split this sentence after “heritable component”, and start “For many genes (…)”.

Line 573: how was this estimated?

Discussion
Line 636: missing the direction of the correlation (positive).

Line 708: “the use”.

Figures
The figures are quite hard to see as the quality is not very good. I recommend improving this issue before publication.

Figure 1A: which divergence did you use in this plot, human-chimp or human-mouse?
Figure 3A: which statistical test was used?
Figure 4B: what do the colours represent?

---

## Round 0.2 · accepted · Accept

Congratulations on addressing the reviewers' comments, and the acceptance of your paper.